# Comparative Transcriptomic and Metabolic Analyses Reveal the Coordinated Mechanisms in *Pinus koraiensis* under Different Light Stress Conditions

**DOI:** 10.3390/ijms23179556

**Published:** 2022-08-23

**Authors:** Yuxi Li, Xinxin Zhang, Kewei Cai, Qinhui Zhang, Luping Jiang, Hanxi Li, Yuzhe Lv, Guanzheng Qu, Xiyang Zhao

**Affiliations:** 1State Key Laboratory of Tree Genetics and Breeding, Northeast Forestry University, Harbin 150040, China; 2Jilin Provincial Key Laboratory of Tree and Grass Genetics and Breeding, College of Forestry and Grassland Science, Jilin Agricultural University, Changchun 130118, China; 3Jilin Province Forestry Survey Design & Research Institute, Changchun 130022, China

**Keywords:** hormone, photosynthesis, transcriptome, metabolome, light stress, *Pinus koraiensis*

## Abstract

Light is one of the most important environmental cues that affects plant development and regulates its behavior. Light stress directly inhibits physiological responses and plant tissue development and even induces mortality in plants. Korean pine (*Pinus koraiensis*) is an evergreen conifer species widely planted in northeast China that has important economic and ecological value. However, the effects of light stress on the growth and development of Korean pine are still unclear. In this study, the effects of different shading conditions on physiological indices, molecular mechanisms and metabolites of Korean pine were explored. The results showed that auxin, gibberellin and abscisic acid were significantly increased under all shading conditions compared with the control. The contents of chlorophyll a, chlorophyll b, total chlorophyll and carotenoid also increased as the shading degree increased. Moreover, a total of 8556, 3751 and 6990 differentially expressed genes (DEGs) were found between the control and HS (heavy shade), control and LS (light shade), LS vs. HS, respectively. Notably, most DEGs were assigned to pathways of phytohormone signaling, photosynthesis, carotenoid and flavonoid biosynthesis under light stress. The transcription factors MYB-related, AP2-ERF and bHLH specifically increased expression during light stress. A total of 911 metabolites were identified, and 243 differentially accumulated metabolites (DAMs) were detected, among which flavonoid biosynthesis (naringenin chalcone, dihydrokaempferol and kaempferol) metabolites were significantly different under light stress. These results will provide a theoretical basis for the response of *P. koraiensis* to different light stresses.

## 1. Introduction

Korean pine (*Pinus koraiensis* Sieb. et Zucc.) is an evergreen coniferous tree belonging to the *Pinus* genus and the Pinaceae family [1]. It is mainly located in northeastern China, as well as in the far eastern region of Russia and on the Korean Peninsula [2]. Timber of Korean pine is widely used in the manufacture of furniture, bridges, ships and other materials because of its excellent material and strong corrosion resistance. Because pine nuts are rich in oil, phenols, amino acids, etc., they have high edible value [3,4]. Meanwhile, pine nuts contain substances such as pinecone polyphenols and *P. koraiensis* polysaccharides, which have various biomedical applications, including free radical scavenging, antioxidant, anti-inflammatory, antiviral, anticancer and antitumor actions [5,6]. *P. koraiensis* was listed as a second-class nationally protected plant in the Chinese Red Data Book in 1999 [1]. In recent years, the natural forest of Korean pine has been seriously destroyed and its natural growth has declined significantly. It is difficult to restore Korean pine forests due to slow growth and reproduction, late fruit and uneasy natural regeneration [7]. However, as the constructive community of broadleaved/Korean pine forest is the typical climax vegetation in Northeast China, Korean pine has higher ecological value owing to its stable population structure and strong environmental adaptability [8]. Korean pine is a dominant tree species for afforestation in Northeast China.

Plants are capable of perceiving and processing information from both biotic and abiotic surroundings for optimal growth and development. Light is one of the most important environmental cues that affects plant development and regulates its behavior [9]. Following the absorption of light, photoreceptors interact with other signal transduction elements, eventually leading to a number of molecular and morphological responses [10]. The entire life cycle of plants is thus strongly influenced by the ever-changing light environment [11]. Alterations in the quality and quantity of ambient light can trigger the plant’s varied and complex responses [12]. Low light stress caused by shade can inhibit physiological responses such as photosynthesis in plants [13]. Compared with full light, the photosynthetic rate of leaves in shade was lower [14]. At the same time, plants will change their physiological and ecological characteristics to avoid the effects of shade. Under a low light environment, plants will improve their ability to capture photons by increasing the chlorophyll content per unit area, increasing chlorophyll a, chlorophyll b and carotenoids, thereby helping plants better utilize light [15]. Light is essential for normal plant growth and development not only as a source of energy but also as a stimulus which regulates numerous developmental and metabolic processes [12]. Numerous studies have documented the occurrence of relationship between light and phytohormones [16]. Plant hormones such as auxin (IAA), cytokinin (CTK), gibberellin (GA), ethylene (ETH) and brassinosteroid (BR) have been known for many years to play a role in light regulation development [17]. In addition, abscisic acid (ABA) also plays an essential role for plants in responses to environmental stresses [18]. Shading conditions can lead to a significant increase in auxin levels [19].

The importance of *P. koraiensis* has resulted in it being the focus of many studies. However, various researchers have mainly analyzed the growth traits, flowering characteristics, timber properties, nutritional components of seeds, genetic diversity and methods used to improve *P. koraiensis* [20,21,22,23,24]. However, few researches have been done on the effect of shading in *P. koraiensis*. Furthermore, many studies mainly focus on physiological responses to shade [25,26], but little is known about the effects of shade stress on the transcriptome and metabolome. Nowadays, with development and application of multi-omics and high-throughput genome sequencing technologies, the combination of transcriptomics and metabolomics provides a high-throughput method to identify the function of the genes involved in metabolic pathways. Studies on the transcriptome of Korean pine have been reported [27]. Esccandon et al. have studied the metabolome of high temperature responses in *Pinus radiata* [28]. These provide a basis for our research. The aim of this study was to investigate the effects of different shading conditions on the physiology, molecular activity and metabolism of *P. koraiensis*. A greater understanding can provide a theoretical basis for the response of *P. koraiensis* to different shading conditions.

## 2. Results

### 2.1. Plant Endogenous Hormone Concentration

To reveal the role of hormones in *P. koraiensis* under shading conditions, we monitored the changes in hormones. The levels of IAA, GA, ABA, SA, CTK and BR were significantly increased in all shading conditions compared with those in the control (Figure 1). The concentration of JA was significantly decreased under all shading conditions compared with the control. The ETH content was not significantly different among the treatments.

### 2.2. Photosynthesis Parameters and Pigment Content

To further validate the effects on photosynthesis under different shading conditions, we checked the physiological parameters of photosynthesis. The results revealed that all shade treatments induced a significant decrease in the Pn compared with the control, and Pn decreased gradually with increasing shading degree. Meanwhile, the results of Gs, Ci, Tr and WUE indicated significant differences among the different treatments (Figure 2).

The species and quantities of chlorophyll in plants changed under different survival environments. Compared with the control, chlorophyll a, chlorophyll b, total chlorophyll and carotenoid were significantly increased under shading conditions (Figure 3A–D). In addition, the chlorophyll a/b ratio under HS was significantly increased compared with that under the other treatments (Figure 3E). The chlorophyll/carotenoid ratio was not significantly different (Figure 3F).

### 2.3. RNA Sequencing and De Novo Assembly

To investigate the changes in gene expression patterns under shading treatment, RNA-seq was performed under different shading conditions. Nine cDNA libraries were constructed using RNA extracted from *P. koraiensis* needles, which were exposed to different shading for HS, LS and CK treatments, three replicates of each treatment. The results of principal component analysis (PCA) revealed that the biological replicates of each sample clustered together were significantly separated from other samples (Figure 4A). A total of 86.32 Gb of clean data were obtained from nine samples, and more than 6.00 Gb of clean data were collected from each sample. The average fragments scoring Q20 and Q30 were over 98% and 94%, respectively, and the average GC content ranged from 44.88% to 46.03% (Appendix A). After filtering out low-quality reads, a total of 575,458,690 clean reads were obtained, which were de novo assembled into contigs by Trinity [29] software (v2.11.0). Moreover, an average of 88.43% clean reads derived from the sequencing samples were mapped to the assembled sequences. Through evaluation of assembly quality according to BUSCO software, a total of 1614 genes were tested and coverage reached 91.90% (1482). These results indicate a high level of assembly quality (Appendix A). A total of 169,624 unigenes were obtained after assembly, with an average length of 837 bp and an Ex90N50 length of 1960 bp.

A total of 169,624 reference transcriptome were aligned with sequences from common databases, including the Nr (nonredundant protein sequences) (https://www.ncbi.nlm.nih.gov/refseq/, accessed on 15 March 2022) [30], Swiss-Prot (a manually annotated and reviewed protein sequence database) (https://www.expasy.org/resources/uniprotkb-swiss-prot, accessed on 15 March 2022) [31], GO (Gene Ontology) (http://www.geneontology.org, accessed on 15 March 2022) [32], KOG (eukaryotic Orthologous Groups) [33], KEGG (Koyoto Encyclopedia of Genes and Genomes) (https://www.genome.jp/kegg/, accessed on 15 March 2022) [34], Pfam (Protein family) (http://pfam.xfam.org/, accessed on 15 March 2022) [35] and Trembl databases, using BLAST for further functional annotations (Figure 4). In total, 79,318 (46.76%) unigenes were matched to a sequence in at least one database above. The number of unigenes identified in the Nr database was the largest, accounting for 44.41% (75,329) of the total unigenes, followed by the TrEMBL (74,197, 43.74%), GO (63,525, 37.45%), KEGG (51,949, 30.63%), SwissProt (51,020, 30.08%), Pfam (50,187, 29.59%) and KOG (41,321, 24.36%) databases. According to a search of the GO database, a total of 59 terms were concentrated, including 28 “biological process”, 18 “cellular component” and 13 “molecular function” (Figure 4B) (Appendix A). Through alignment of unigene sequences with Nr database using DIAMOND [36] BLASTX software, the unigene sequences had the strongest matches with the gene sequences from *Picea sitchensis* (38.85%), and similarities to sequences from other species were also observed (Figure 4D).

### 2.4. Annotations and Enrichment Analysis of Differentially Expressed Genes (DEGs)

DEseq [37,38] was used to identify DEGs in different shading treatments with specified thresholds. DEGs exist in all comparison groups, and the results shown in Figure 5A,B. To further explore the biological function of DEGs, functional classification and enrichment analysis of the identified DEGs was carried out using the GO annotation and KEGG pathway system in the three comparison groups. The GO functional classification mainly included biological processes, cellular components and molecular functions. In the three comparison groups, the terms associated with biological process were the most enriched ones, which mainly included cell process, metabolic process and response to stimulus; however, the terms of molecular function were the lowest, which mainly included binding, catalytic activity and transporter activity. For the cellular component, DEGs mainly annotated cell and cell part (Appendix A).

Figure 5C–E shows the top 20 enriched pathways in CK vs. HS (Appendix A), CK vs. LS (Appendix A), and LS vs. HS (Appendix A). According to the KEGG pathway analysis, a total of 135 pathways were enriched in CK vs. HS, with biosynthesis of galactose metabolism being the most enriched pathway, followed by phenylpropanoid biosynthesis and flavone and flavonol biosynthesis (*p* value < 0.05). For CK vs. LS, the analysis indicated that plant–pathogen interaction, protein processing in endoplasmic reticulum and biosynthesis of secondary metabolites were significantly enriched (*p* value < 0.05). Furthermore, cutin, suberin and wax biosynthesis and protein processing in the endoplasmic reticulum were significantly enriched (*p* value < 0.05). These results showed a complex regulatory network of *P. koraiensis* for shading treatment.

### 2.5. Analysis of Transcriptome Factors (TFs)

Transcription factors (TFs) can read the genetic code by binding to cis-regulatory elements to activate or repress gene expression. TFs play a key role in plant development and gene expression regulation, forming complex gene regulatory networks [39,40]. To explore the TFs in response to shading in *P. koraiensis*, a total of 331 TF genes were identified as DEGs and encoded 68 TF families (Appendix A). As shown in Figure 6, the top four TF families in DEGs were MYB-related (11%, 38 genes), AP2-ERF (10%, 33 genes), bHLH (10%, 32 genes) and HB (8%, 26 genes) (Appendix A). 

### 2.6. Differentially Accumumulated Metabolite (DAM) Profiling

To further explore the changed metabolites of *P. koraiensis* in shade, all samples were identified to metabolome profiling via untargeted LC-MS. The PCA of the three samples revealed effective separation (Figure 7A). In total, 911 metabolites were identified, 243 of which were differentially accumulated metabolites (DAMs) (Figure 7B). A Venn diagram shows the number of significantly different metabolites among the three comparisons. In CK vs. HS and CK vs. LS, a total of 158 DAMs (58 upregulated and 100 downregulated) and 62 DAMs (24 upregulated and 38 downregulated) were identified. In total, 175 DAMs with 80 upregulated and 95 downregulated metabolites were identified in LS vs. HS. Three comparisons revealed 13 common DAMs (Figure 7C). The number of DAMs in CK vs. HS was greater than that in CK vs. HS, which may be caused by different shading levels. This result was similar to the DEG results. A total of 243 DAMs were divided into 10 categories and the most abundant categories were flavonoids (34%), followed by phenolic acids (16%) and lignans and coumarins (12%).

To understand the biological function of DAMs, we used the KEGG pathway database to examine the DAMs-associated pathways. The top 20 enriched pathways of annotated DAMs across each comparison were further analyzed. A total of 57 pathways were enriched in CK vs. HS (Figure 7D). The top four enriched pathways were arginine biosynthesis, plant hormone signal transduction, biosynthesis of secondary metabolites and phenylpropanoid biosynthesis (*p* value < 0.05) (Appendix A). In CK vs. LS, flavone and flavanol biosynthesis and phenylpropanoid biosynthesis were significantly enriched (*p* value < 0.05) (Figure 7E) (Appendix A). The analysis indicated that arginine biosynthesis, plant hormone signal transduction and purine metabolism were significantly enriched (*p* value < 0.05) in LS vs. HS (Figure 7F) (Appendix A). The plant hormone signal transduction pathway was significantly enriched in both comparisons, so we conjectured that it plays an important role in *P. koraiensis* under different shading conditions.

### 2.7. DEGs Related to Hormone Signaling

Hormones have been proposed to be essential regulators in plant development. We further investigated DEG regulation in multiple hormone signal transduction pathways [41] in this study. The results revealed a total of 165 DEGs encoding 27 enzymes in eight hormone signaling transduction pathways that were regulated under different shading conditions (Figure 8) (Appendix A). In the auxin signal transduction pathway, most genes encoding AUX1 (one gene), ARF (ten genes), GH3 (two genes), AUX/IAA (two genes) and SAUR (15 genes) enzymes were more highly expressed in shade treatments than in the control. There were 16 genes encoding three enzymes mainly involved in the zeatin signal transduction pathway. All B-ARR genes were upregulated in all shade treatments, and the majority of CRE1 and AHP genes were upregulated in shade. In the gibberellin signal transduction pathway, the transcript abundance of DELLA (three genes) was higher in HS and LS than in CK, and the majority of GID1, GID2 and TF genes were upregulated in HS. A total of 11 genes encoding four enzymes in the abscisic acid signal transduction pathway, and the PYR/PYL genes were upregulated in HS. All genes of ABF and PP2C were upregulated in all shade treatments, and HS induced a greater increase than LS. In the ethylene signal transduction pathway, a total of five genes were differently expressed in different treatments. In the brassinosteroid signal transduction pathway, most of the genes encoding BAK1, BRI1 and BKI1 were upregulated under shade conditions. In the jasmonic acid signal transduction pathway, all MYC2 genes except Cluster-27241.104920 were upregulated in all shade treatments, and the HS treatment induced a greater increase than LS treatment. In the salicylic acid signal transduction pathway, a total of five DEGs (two genes downregulated in LS and HS compared with CK, three genes upregulated in LS and HS compared with CK) encoded PR-1. The network map of light and all hormone signal transduction pathways was complex and was regulated by many genes.

### 2.8. DEGs Related to Photosynthesis

Light harvesting is the first step in the process of photosynthesis, and is regulated by physiological status and environmental signals [42]. First, DEGs in the photosynthesis and photosynthesis-antenna protein pathways [43] were identified in each comparison group (Appendix A). A total of 29 DEGs related to the photosynthesis process were collected under different shade treatments (Figure 9A). Furthermore, most of the photosynthesis Ⅰ reaction center subunit-related genes PsaN, PsaO, PsaG and PsaC were upregulated in shade. In addition, the photosynthesis Ⅱ core complex protein-related genes PsbA, PsbB, PsbC, PsbS, PsbQ and PsbZ showed the opposite trend. Other photosynthesis-related genes including photosynthesis electron transport genes PetJ, PetF, PetH and F-type ATPase synthesis-related genes atpA, atpB, atpF, atpE and atpH were also differentially expressed in the shade treatments compared with the CK treatment. In the photosynthesis-antenna protein pathway, a total of 16 genes were differentially expressed (Figure 9B). Furthermore, the light-harvesting complex chlorophyll protein complex genes Lhca1, Lhcb1, Lhcb2 and Lhcb7 were differentially expressed. The Lhca1 gene was increased with reduced light intensity. On the other hand, the Lhcb7 gene showed the opposite pattern. Most Lhcb1 and Lhcb2 genes were significantly upregulated in the HS treatment. These results suggested that *P. koraiensis* may cope with shading stress by regulating the expression of photosynthesis-related genes.

### 2.9. DEGs Related to Carotenoid and Chlorophyll Biosynthesis

Chlorophyll biosynthesis [44] begins with glycine and the ALAS gene product that converts glycine to 5-aminolevulinare. As shown in Figure 10A, a total of 28 DEGs were annotated as key genes encoding enzymes related to porphyrin and chlorophyll metabolism (Appendix A). The DEGs of SGR, RCCR and POR were more highly expressed under shading treatments than in the control treatment. The majority of CHL genes were downregulated in shade. Furthermore, a total of seven DEGs were encoded in PPD, three genes were upregulated in HS and four genes upregulated in LS.

Carotenoids are essential for functional photosynthesis [45]. The results showed that a total 17 DEGs were annotated as key genes encoding enzymes related to carotenoid biosynthesis [46] (Figure 10B) (Appendix A). All DEGs of PSY, LUT1, LUT5 and VDE were downregulated in the shade treatment compared with the control. All HYD genes were upregulated in HS. The majority of ZEP genes were upregulated under shading condition compared with the control.

### 2.10. Flavonoid Concentrations Induced by Shade Treatments

The flavonoid biosynthesis pathway [47] was enriched in all comparison groups. Enrichment analysis showed that there were 73 DEGs encoding nine enzymes in the flavonoid biosynthesis pathway that were differentially expressed (Figure 11) (Appendix A). The majority of CHS, ANS and F3H genes were downregulated in shade treatments. Simultaneously, flavonoids were the most abundant categories in all DAMs, and a total of 82 metabolites involved eight classes of flavonoids. These eight classes of flavonoids were chalcones, flavanols, flavanones, flavanonols, flavones, flavonoid carbonoside, flavonols and isoflavones. There were six DAMs in different treatments (Figure 11). The DAMs naringenin, naringenin chalcone and dihydromyricetin accumulated at lower levels in HS and LS treatments than in CK treatment. Catechin and dihydrokaempferol accumulated the highest levels in LS. 

### 2.11. qRT-PCR Validation

To evaluate the reliability of the sequencing results, a total of nine genes were selected to confirm the RNA-seq data. These genes were selected for expression analysis in CK, LS and HS by using qRT–PCR. The relative expression of these candidate genes was similar to the RNA-seq results, and the results were firmly validated (Appendix A). 

## 3. Discussion

*P. koraiensis*, an important conifer in Northeast China, plays a key role in the succession and development of forests. Light intensity is one of the most important environmental factors in forest regeneration [48]. Therefore, it is particularly important to explore the effect of different light intensities on *P. koraiensis* for understory regeneration. Nevertheless, few studies on *P. koraiensis* under shade at present. Here, we simulated understory conditions using shade nets and compared the effects of different shade treatments with the regulation of physiologies and molecular mechanism. Further, combined with physiology, transcriptomic and metabolomic, we found that *P. koraiensis* responds to shade through the regulation of physiological indicators, genes and metabolite accumulation.

### 3.1. Effect of Artificial Shading on the Plant Hormonal Regulation of P. koraiensis 

In recent years, knowledge of plant hormone signaling transduction pathways has increased rapidly, and receptors are now known for all major hormones. Many light responses in plants are mediated through changes in hormonal metabolism and distribution, and the links between light perception and hormonal regulation are being elucidated for many processes [49]. Our analysis of gene regulation in multiple plant hormone signaling pathways revealed that the growth-promoting hormones IAA, GA, ABA, JA, SA, BR and CTK were all induced under different shade treatments. At the same time, genes of eight hormone signaling transduction pathways were induced. This could be a crosstalk from a hormonal cascade signaling network [50,51]. IAA, GA and ABA increased in the HS and LS treatments in our study, which showed a mechanism of shade response. In addition to being shown in *Arabidopsis* [52], the key role of auxin in shade has been confirmed in crop species [53] such as potato [54] and peanut [14]. Additionally, auxin biosynthesis and transport and auxin sensitivity were also enhanced under shade [55,56]. The expression of auxin-responsive genes is also dramatically affected by shade treatment [57,58]. Gibberellins also function as growth promoters, which play an important role in shade. Increasing gibberellin concentration in *Arabidopsis* [59], bean [60] and cowpea [61] in response to shade has been documented. ABA responds to a variety of environmental stresses. Shade conditions increase ABA levels in *Helianthus annuus* [62] and *Solanum lycopersium* [63]. JA and SA have been widely regarded as key signaling molecules that regulate the plant stress response and defense [64,65]. The antagonism between JA and SA has been reported [66]. In this study, IAA, GA, ABA, SA, BR and CTK contents were significantly increased, whereas most DEGs related to these hormone signal transduction pathways were upregulated. As they showed similar trends, we suggested that these hormone contents changed due to the DEGs induced. Many of the hormone response pathways seem to be interconnected with each other and with light signaling [17]. Meanwhile, various phytohormones are involved and coordinate to shape shade-regulated plant architecture. Changes in the synthesis, transportation and signaling of these hormones occur via hormone levels and the expression of these hormone-associated transcription factors.

### 3.2. Effect of Artificial Shading on the Plant Photosynthetic Regulation of P. koraiensis 

Photosynthesis is the basic energy conversion process. Light is not only the basic driving force of photosynthesis but is also an important stress factor [67]. Our study revealed a decrease in Pn with decreasing light intensity. The lower irradiance in the shade limits photosynthesis [68]. The light-induced decline in photosynthetic activity mainly affects the photosynthesis Ⅱ (PSⅡ) complex [67,69]. We found that the DEGs related to PSII were downregulated in shade, possibly due to photoinactivation of PSII, which caused the Pn to decrease. In particular, psbS DEGs were all downregulated with increasing shade degree, and they were consistent with the Pn data. Therefore, we conjectured that psbS plays a key role in photosynthesis under shade. Meanwhile, the majority of DEGs related to the light-harvesting complex chlorophyll a/b binding protein were upregulated, suggesting an attempt to obtain more light in shade. In addition, a large number of studies on physiological and molecular genetic have indicated that plants possess distinct photoreceptors [10]. Photoreceptors are involved in perceiving the differences between full light and shade [68]. Phytochrome is a type of photoreceptors, which could recognize different light information including intensity, wavelength, duration and direction [70]. There are five members of the phytochrome gene family designated as phytochrome A (PHYA), PHYB, PHYC, PHYD and PHYE [71]. In *Arabidopsis*, PHYB plays a dominant role in the shade avoidance response, and PHYD and PHYE play minor roles [72]. In shade conditions, PHYs (mainly PHYB) are inactivated, leading to accumulation of PIF3, PIF4 and PIF5, and activation of PIF7 through dephosphorylation [73,74,75]. Ultimately, they promote elongation growth [76]. However, we have little knowledge about the phytochrome regulation in the shade avoidance response of *P. koraiensis*. These requires further study in future research.

### 3.3. Effect of Artificial Shading on the Plant Photosynthetic Pigments Regulation of P. koraiensis

Light harvesting is the first step in photosynthesis; therefore, the light-harvesting antennae must be regulated in response to their physiological status and environmental signals [77]. Chlorophylls ligated to light-harvesting complex (LHC) proteins and carotenoids mainly serve as light-harvesting antennae in higher plants [78]. The biogenesis of LHC involves the synthesis and assembly of chlorophyll, carotenoids and apoproteins [79]. In this study, chlorophyll a, chlorophyll b and total chlorophyll were significantly increased under shading conditions, which is consistent with the results of Adrian et al. [80]. Chlorophyll is the most abundant tetrapyrrole in plants, functioning as a pigment for light energy harvesting and transfer to the reaction center in photosynthesis [44]. Chlorophyll metabolism is a highly coordinated process that is executed via a series of cooperative reactions catalyzed by numerous enzymes [81]. In the chlorophyll biosynthesis pathway, we found that the expression of four POR DEGs was upregulated in shade. This result was consistent with the results of previous physiological experiments. POR occupies the most downstream position in the chlorophyll biosynthesis pathway and encodes an enzyme that catalyzes the formation of chlorophyllide a from protochlorophyllide [82]. Sakura et al. found that POR is a critical gene in chlorophyll biosynthesis [83]. Our results confirm the reliability of our conjecture about the key genes in chlorophyll biosynthesis under shade. Carotenoids play an essential role in photosynthesis, photomorphogenesis and plant development. Carotenoids serve as accessory pigments to harvest light for photosynthesis and constitute the basic structural units of the photosynthesis apparatus [46]. Additionally, carotenoids have been implicated in the interactions of plants with their environment [15]. In this study, the carotenoid content was increased in shade, and we found that multiple genes were reduced. In particular, most ZEP genes were upregulated in shade. Further epoxidation of zeaxanthin by zeaxanthin (ZEP) epoxidase produces violaxanthin, and this reaction allows plants to adapt to low light stress [46]. Therefore, plants acclimated to shade conditions adapt to the low-light environment by increasing the ratio of chlorophyll and carotenoids.

### 3.4. Effect of Artificial Shading on the Plant Flavone Metabolism Regulation of P. koraiensis

Plants can undergo physiological adjustments in response to a wide range of environmental stimuli. These mechanisms may include activation of flavonoid biosynthesis [84]. Flavonoids are a group of phenolic secondary metabolites that are widespread among plants and are involved in many plant functions. Flavonoids are frequently activated in response to a variety of abiotic stressors, such as drought [85], cold [86] and UV-B exposure [87]. Chemical composition analyses in the present study revealed a decrease in the accumulation of multiple flavonoids in the shade. We found that CHS, CHI, F3H, F3′H, FLS, F3′5′H, ANS and LAR genes were all significantly differentially expressed in different treatments. The biosynthesis of the flavone backbone originates from the general phenylpropanoid pathway followed by the flavonoid biosynthesis branch [88]. CHS and CHI are upstream of the other genes in the pathway. Chalcone synthase (CHS) is positioned at the entry point of the pathway [89]. The majority of CHS and CHI genes were downregulated in shade. These results are consistent with the flavonoid data. In summary, CHS and CHI both played a key role in flavonoid biosynthesis under shade. Flavonoids are thought to participate in the protection of the photosynthesis apparatus against photoinhibition under excessive light [90]. In our study, naringenin, naringenin chalcone and dihydromyricetin were downregulated in shade. Shade can cause a reduction in the concentration of flavonoids in tea [47]. This finding is consistent with our study. Flavonoids exhibited a degree of plasticity in shade [91]. These results suggest that shade treatment was effective in reducing the biosynthesis of flavonoids. Shade can mitigate the damage of photoinhibition in *P. koraiensis*. This indicates that shade avoidance may share some common regulatory mechanisms with defense and immunity to other stresses.

## 4. Materials and Methods

### 4.1. Plant Materials and Shading Conditions

The natural distribution range of *P. koraiensis* is mainly in Northeast China. Four-year-old seedlings of *P. koraiensis* were cultivated at the experimental field of Daquanzi Forest Farm in Harbin City, Heilongjiang Province (126°55′–128°19′, 45°30′–46°01′). A total of two different shading treatments and control were used, including light shade (LS, 80% light), heavy shade (HS, 20% light) and control (CK, 100% light). For the shading treatments, light was artificially reduced using black nets covering all seedlings and allowing 80% and 20% of the photosynthetic photon flux density (PPFD), and PPFD measured using the portable CIRAS-3 photosynthesis system (PP Systems, Amesbury, MA, USA). There were 50 seedlings per treatment and they were not occluded with each other. After shading for 45 days, we collected samples. Each sample consisted of leaves from three plants grown in the same shade treatment. All leaf samples were immediately frozen in liquid nitrogen and stored at −80 °C.

### 4.2. Plant Endogenous Hormone Measuremens

With the development and application of the constructive metabolism of plant endogenous hormones and the isolation and identification of signaling mutants, researchers have begun to understand the effect of plant hormone synthesis, transport signal transduction and degradation on plants. The contents of auxin (IAA), gibberellin (GA), abscisic acid (ABA), ethylene (ETH), jasmonic acid (JA), salicylic acid (SA), brassinosteroid (BR) and cytokinin (CTK) were measured using Enzyme-Linked Immunosorbent Assays (ELISA). The samples were random collected from three seedlings and stored at −80 °C for phytohormone content analyse. The method of phytohormone extraction followed the description by Zhang et al. [92] with few modifications. The samples were ground with liquid nitrogen, and 5 mL 80% methanol (containing 10 mg/L butylhydroxytoluene) was added to the 0.5 g sample powder. Then, hormones were extracted at 4 °C for 4 h. We absorbed the supernatant into a new centrifuge tube after centrifugation (5000× *g* at 4 °C for 10 min). The residues were extracted again according to the above steps. The merged supernatants were dried at 35 °C under nitrogen gas. Finally, the extracts were diluted with 80% methanol as the samples to be tested. The ELISA kit was supported by Shanghai Enzymatic Biotechnology Company Ltd. The specific operation steps were done according to the manufacturer’s instructions.

### 4.3. Photosynthesis Measurements

Photosynthesis parameters were measured using the portable CIRAS-3 photosynthesis system (PP Systems, Amesbury, MA, USA). These parameters include the net photosynthesis rate (Pn), stomatal conductance (Gs), intercellular CO_2_ concentration (Ci), transpiration rate (Tr) and water use efficiency (WUE). The determination time was between 8:30 and 11:00 am. on sunny and clear days. Light was provided with ambient lighting during the time of measuring. The CO_2_ concentration was that of the ambient air. Other environmental factors were not specifically controlled. Three replicates were made from three different plants in each treatment. 

### 4.4. Photosynthetic Pigment Determination

Photosynthesis chlorophyll a, chlorophyll b, total chlorophyll and carotenoid contents were determined with the method described by Zhang et al. [93] with few modifications. The leaves of three seedlings were randomly selected and mixed, then frozen in liquid nitrogen immediately. The leaves were ground with liquid nitrogen, 0.1 g samples into new centrifuge tubes were taken. Then, we added 10 mL 95% ethanol into the centrifuge tubes. The pigments were extracted in the dark, and samples were shaken occasionally until blanched. The absorbance values of the supernatant were measured at 470, 649 and 664 nm after centrifugation (5000× *g* for 10 min). The pigment contents were calculated using the previous study [93], seen in the equations as follows:$$Chl\;a=13.36\times A_{664}-5.19\times A_{649}$$
$$Chl\;b=27.43\times A_{649}-8.12\times A_{664}$$
total Chl=Chl a+Chl b
$$Car=\left(1000\times A_{470}-2.13\times Chl\;a-97.64\times Chl\;b\right)/209$$
where *A*_649_, *A*_664_ and *A*_470_ are measured absorbance at 649 nm, 664 nm and 470 nm, respectively.

### 4.5. RNA Extraction and Sequencing

The total RNA from each sample was extracted using a plant total RNA isolation kit (Tiangen, Beijing, China) based on the manufacturer’s instructions. A total of nine cDNA libraries (three replicates each of CK, LS and HS) were constructed. After estimating the quality, concentration and integrity of total RNA, the RNA was sequenced on the Illumina HiSeq platform (HiSeq^TM^ 2500, San Diego, CA, USA). To ensure the quality of the raw reads, they were filtered using Fastp [94] software (version 0.12) to obtain high-quality clean reads. De novo assembly of unigenes was performed using Trinity [29] software (v2.11.0). The longest transcript at each locus was considered a unigene, and the unigene IDs were automatically generated by the software. Gene functional annotation of the assembled unigenes was performed using public databases, including Nr [30], Swiss-Prot [31], GO [32], KOG [33], KEGG [34], Pfam [35] and Trembl databases.

### 4.6. Analysis of Differentially Expressed Genes (DEGs) and Functional Annotation

Gene expression levels for each gene were normalized using DESeq2 [37,38] and summarized as FPKM (Fragments Per Kilobase pair per Million reads) values. We analyze the differentially expressed genes (DEGs) by the DEseq2 R package (1.20.0) and the significant differential expression was |log_2_ Fold Change| ≥ 1 and FDR < 0.05. The GOseq R package was employed for GO function analysis. The KEGG pathways enriched in the DEGs were identified using KOBAS software (2.0).

### 4.7. Quantitative Real-Time Polymerase Chain Reaction (qRT-PCR) Validation

Nine genes were selected from the DEGs and detected using real-time quantitative techniques to verify the correctness of the transcriptome results. The RNA of each sample was extracted using a plant total RNA isolation kit (Tiangen, Beijing, China) based on the manufacturer’s instructions. The cDNA of all samples was obtained using the Prime Script RT reagent Kit with gDNA Eraser (TaKaRa, Kyoto, Japan), and the specific operation steps followed the instructions of the manufacturer. In addition, qRT–PCR was performed on the ABI 7500 Fast Real-Time Detection System by using the TaKaRa SYBR Green Mix kit (TaKaRa, Kyoto, Japan). The PCR amplification experiment was performed with 10 μL 2 × SYBR Green premix ExTaq II, 0.4 μL Rox Reference Dye II, 0.8 μL primer-F/R, 2 μL cDNA and ddH_2_O to 20 μL. Then, qRT–PCR was performed as follows: 95 °C for 30 s, 40 cycles of 95 °C for 5 s and 60 °C for 35 s, 95 °C for 5 s, 60 °C for 1 min and 95 °C for 15 s. The reference gene was 18S-RNA, and all primers are listed in Appendix A. The relative expression analysis of quantitative data was performed using the 2^−ΔΔCT^ method [95].

### 4.8. Metabolite Identification and Functional Analysis

To identify metabolites under different shading conditions, we performed a metabolome analysis of leaves from *P. koraiensis*. Metabolite profiling was performed using a widely targeted metabolome method by Wuhan MetWare Biotechnology Co., Ltd. (Wuhan, China), three replicates for each sample. The methods used were described in previous articles [96,97]. The leaf samples for metabolite extraction were freeze-dried by a freeze-dryer (Scientz-100F) and crushed using a mixed mill (MM400, Retsh) for 1.5 min at 30 Hz. Metabolites were extracted from 0.1 g of each tissue sample with 1.2 mL 70% methanol, and the samples were vortexed 30 s every 30 min and repeated six times. Mixtures were stored at 4 °C for one night. The supernatants were filtered (PTFE, 0.22 μm) after centrifugation (12,000 rpm for 10 min)*,* which were used for UPLC–MS/MS analysis. Samples were analyzed by UPLC–MS/MS using an AB 4500 QTRAP UPLC/MS/MS equipped with a SHIMADZU Nexera X2 unit. Samples of 4 μL were injected onto an Agilent SB-C18 column (2.1 mm × 100 mm, 1.8 μm). The column was held at 40 °C, and the flow rate was controlled at 0.35 mL/min. A gradient using ultra-pure water (containing 0.1% formic acid) as Buffer A and acetonitrile (containing 0.1% formic acid) as Buffer B was used. The main steps were as follow: 5–95% B from 0.0 to 9.0 min, 95% B from 9.0 to 10.0 min, 5–95% B from 10.0–11.1 min and 5% B until 14 min. For mass spectrometry, gas temperature, 550 °C; ion spray voltage, 5500 V/−4500 V; ion source gas Ⅰ, ion source gas Ⅱ, and curtain gas were 50, 60 and 25 psi were used. All chromatograms and mass spectra were analyzed by Analyst software (1.6.3).

Normalized metabolite data were used to compare all samples. Hierarchical cluster analysis (HCA) and orthogonal partial least-squares discriminant analysis (OPLS-DA) were performed to identify the accumulation pattern in metabolites from nine leaf samples of different shading conditions. The OPLS-DA model was used to identify the differences in metabolite composition between the samples by VIP ≥ 1 and fold change ≥ 2 or ≤ 0.5. The metabolites were mapped to KEGG metabolic pathways.

### 4.9. Statistical Analysis

The subsequent statistical analyses were performed using Excel 2021 and SPSS version 26.0 (International Business Machines, Armonk, NY, USA). One-way ANOVA was performed to test the effect of shading degree on physiological parameters, followed by multiple comparisons of means, which were conducted using S-N-K tests at *p* < 0.05 to evaluate the different physiological responses of different shading treatments.

## 5. Conclusions

In this study, we found that the auxin, gibberellin and abscisic acid of *P. koraiensis* were significantly increased under all shading conditions compared with the control. The contents of chlorophyll and carotenoid also increased as the shading degree increased. Furthermore, the molecular regulatory mechanisms in *P. koraiensis* under different shade conditions by combining metabolome and transcriptome data were studied. We found that plant hormone signal transduction, photosynthesis, chlorophyll and carotenoid biosynthesis pathway were enriched under different light stresses. In addition, flavonoids were the major differential metabolites of *P. koraiensis* under light stress. These can be responsible for maintaining the light stress acclimated status via complex regulatory networks. These results provide novel insights into the molecular regulation of *P. koraiensis* under light stress.

## Figures and Tables

**Figure 1 ijms-23-09556-f001:**
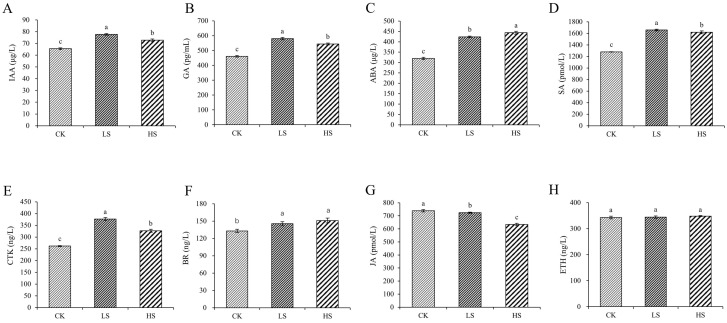
Changes in plant hormone content in *P. koraiensis* under light stress. (**A**) IAA; (**B**) GA; (**C**) ABA; (**D**) SA; (**E**) CTK; (**F**) BR; (**G**) JA; (**H**) ETH. CK: control. LS: light shade. HS: heavy shade. The error bars represent the standard error. Different letters indicate significant differences among different treatments.

**Figure 2 ijms-23-09556-f002:**
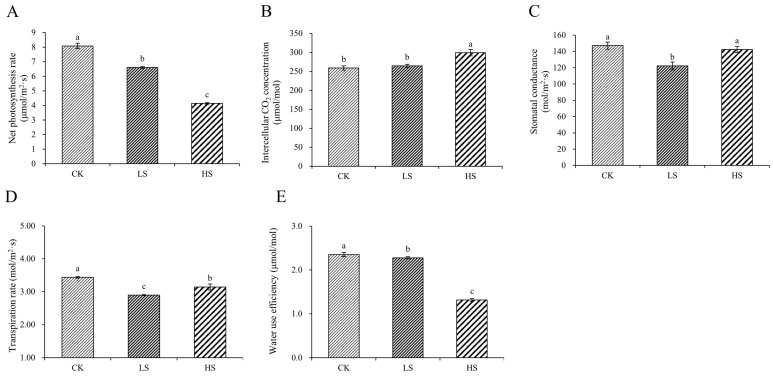
Changes in photosynthetic indicators in *P. koraiensis* under light stress. (**A**) Net photosynthetic rate. (**B**) Intercellular CO_2_ concentration. (**C**) Stomatal conductance. (**D**) Transpiration rate. (**E**) Water use efficiency. CK: control. LS: light shade. HS: heavy shade. The error bars represent the standard error. Different letters indicate significant differences among different treatments.

**Figure 3 ijms-23-09556-f003:**
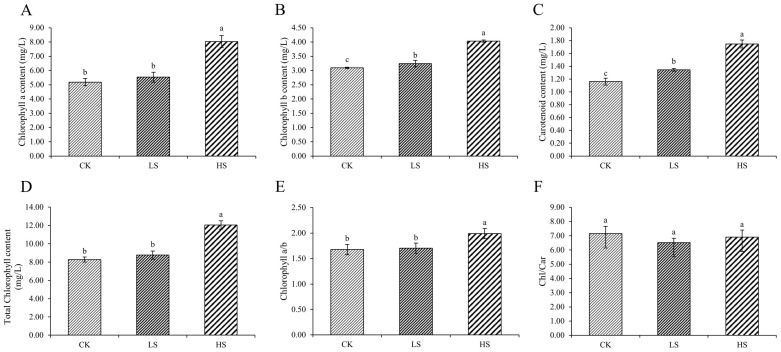
Changes in pigments in *P. koraiensis* under light stress. (**A**) Chlorophyll a content. (**B**) Chlorophyll b content. (**C**) Total chlorophyll content. (**D**) Carotenoid content. (**E**) Chlorophyll a/b. (**F**) Chlorophyll/Carotenoid. CK: control. LS: light shade. HS: heavy shade. The error bars represent the standard error. Different letters indicate significant differences among different treatments.

**Figure 4 ijms-23-09556-f004:**
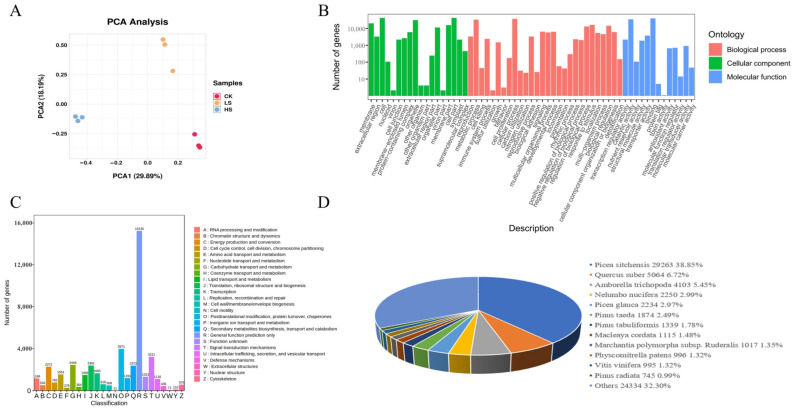
Functional annotation of assembly unigenes in *P. koraiensis* under light stress. (**A**) PCA score plot of expression profiles from different samples. CK: control. LS: light shade. HS: heavy shade. (**B**) GO classifications of assembly unigenes. (**C**) KOG functional classification of assembly unigenes. (**D**) Similarity of *P. koraiensis* sequences with other species in Nr.

**Figure 5 ijms-23-09556-f005:**
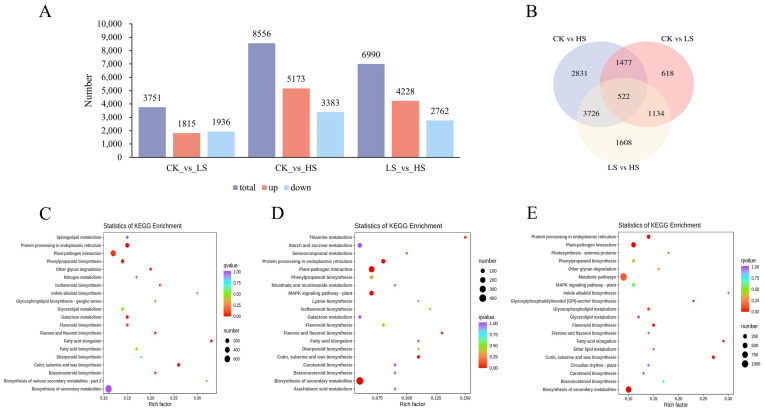
Identification and functional enrichment of differentially expressed genes of *P. koraiensis* under light stress. (**A**) The statistics of upregulated and downregulated DEGs in different comparison groups. (**B**) Venn diagram of DEGs in different comparison groups. (**C**) The top 20 KEGG enrichment pathways of DEGs in CK vs. HS. (**D**) The top 20 KEGG enrichment pathways of DEGs in CK vs. LS. (**E**) The top 20 KEGG enrichment pathways of DEGs in LS vs. HS. CK: control. LS: light shade. HS: heavy shade.

**Figure 6 ijms-23-09556-f006:**
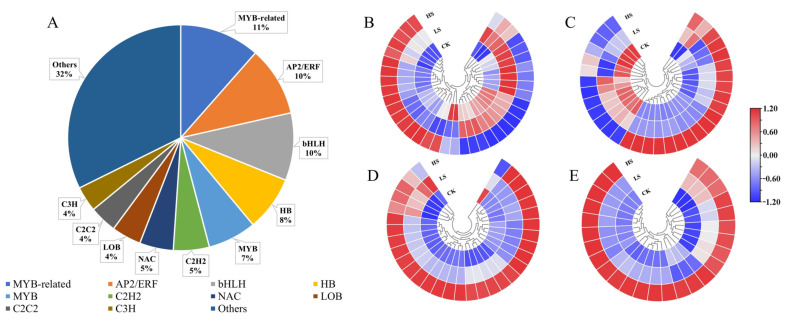
Changes in the expression levels of DEGs encoding transcription factors of *P. koraiensis* under light stress. (**A**) The percentage of transcription factor families in all comparison groups; Heatmap of DEGs involved in (**B**) MYB-related. (**C**) AP2/ERF. (**D**) bHLH and (**E**) HB transcription factor family. CK: control. LS: light shade. HS: heavy shade. The color scale from blue to red indicates the expression value from low to high.

**Figure 7 ijms-23-09556-f007:**
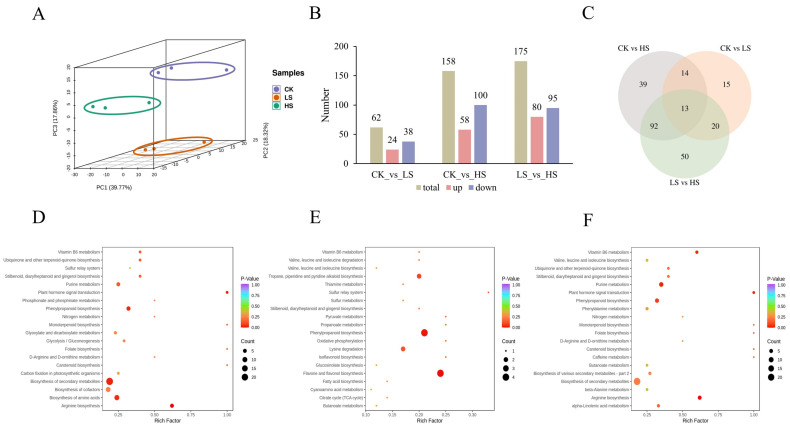
Identification of differentially abundant metabolites in *P. koraiensis* under light stress. (**A**) PCA score plot of expression profiles from different samples. (**B**) The statistical analysis of upregulated and downregulated differentially expressed metabolites in different comparison groups. (**C**) Venn diagram showing the number of metabolites in CK, LS and HS. (**D**) The top 20 KEGG enrichment pathways of differentially accumulated metabolites (DAMs) in CK vs. HS. (**E**) The top 20 KEGG enrichment pathways of DAMs in CK vs. LS. (**F**) The top 20 KEGG enrichment pathways of DAMs in LS vs. HS. CK: control. LS: light shade. HS: heavy shade.

**Figure 8 ijms-23-09556-f008:**
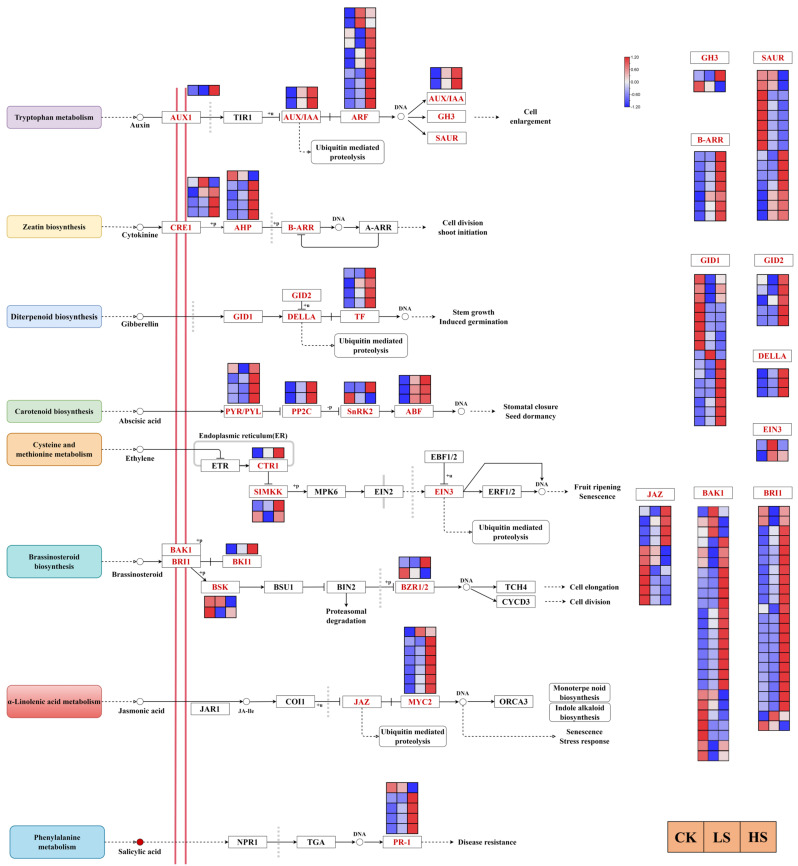
Analysis of DEGs related to phytohormone signaling pathways. The gene expression levels in three samples are shown in different columns in the colored boxes, and different rows represent different genes. CK: control. LS: light shade. HS: heavy shade. The color scale from Min (blue) to Max (red) refers to the expression value from low to high.

**Figure 9 ijms-23-09556-f009:**
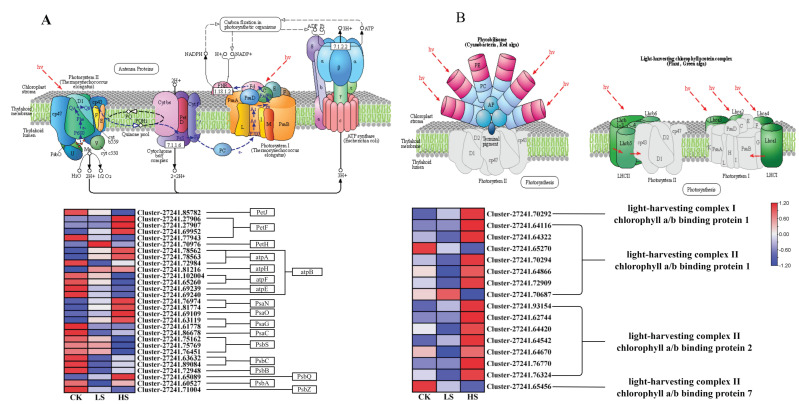
Expression of DEGs involved in response to light and photosynthesis in *P. koraiensis* under light stress. (**A**) The structure and mechanism of photosynthesis and a heatmap of the expression levels of DEGs involved in photosynthesis. (**B**) The structure of photosynthesis-antenna proteins and a heatmap of the expression levels of DEGs that were found to be involved in response to light stimuli. CK: control. LS: light shade. HS: heavy shade. The color scale from Min (blue) to Max (red) refers to the expression value from low to high.

**Figure 10 ijms-23-09556-f010:**
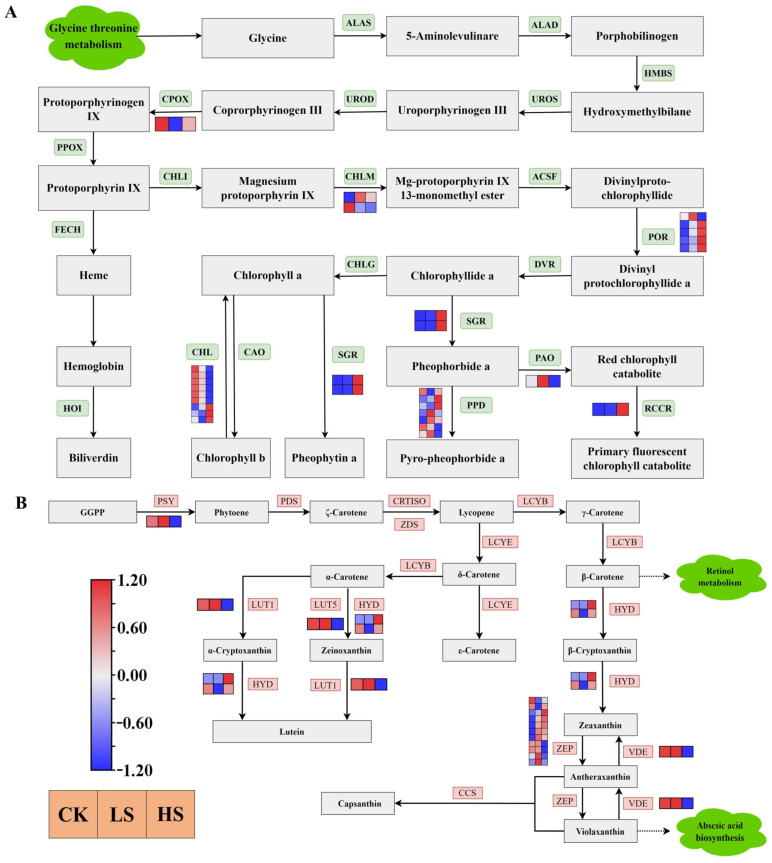
Transcriptional profiling of differentially expressed genes (DEGs) associated with pigment metabolism in *P. koraiensis* under light stress. (**A**) The DEGs involved in chlorophyll metabolism. (**B**) The DEGs involved in carotenoid biosynthesis. CK: control. LS: light shade. HS: heavy shade. The color scale from Min (blue) to Max (red) refers to the expression value from low to high.

**Figure 11 ijms-23-09556-f011:**
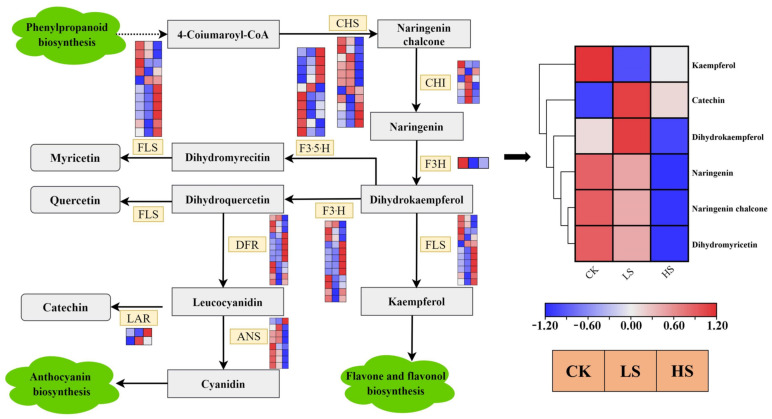
The expression of genes in the flavonoid biosynthetic pathways in the *P. koraiensis* under light stress. CHS chalcone synthase, CHI chalcone isomerase, F3H flavanone 3-hydroxylase, F3′5′H flavonoid 3′,5′-hydroxylase, LAR leucocyanidin reductase, F3′H flavonoid 3′-hydroxylase, FLS flavonol synthase, DFR dihydroflavonol 4-reductase, ANS anthocyanidin synthase. Reconstruction of the flavonoid biosynthetic pathway with the differentially expressed structural genes and the differentially accumulated metabolites (DAMs) in the flavonoid biosynthetic pathway. The cluster marker on the right side of the heatmap represents the names of each metabolite. CK: control. LS: light shade. HS: heavy shade. The color scale from Min (blue) to Max (red) refers to the metabolite contents from low to high.

## Data Availability

In this study, data from three samples were deposited in NCBI under the accession number PRJNA848848.

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
