# Peer review of "Comparative Transcriptomic and Metabolic Analyses Reveal the Coordinated Mechanisms in Pinus koraiensis under Different Light Stress Conditions"

_ijms, 2022, doi:10.3390/ijms23179556_

Round 1

Reviewer 1 Report

This study described the physiological, transcriptomic, and metabolite changes for Pinus koraiensis under different light stress conditions. Experiments were designed, performed and presented well. The abundant results and careful analyses can deepen our knowledge about plant physiology under shade and light in general. 

However, I have two major concerns about some figures and methods.  1) Did you use the published graphs or pathways in the figure 8,9,10,11? If so, please cite the original references. Additionally, please check the journal publication guidelines about how to use the published graphs from other studies.  I don’t think authors can use the published photos without the original authors’ agreement.  2) The methods lack details. It is expected that methods would look like a “cooking recipe”, so readers can replicate the procedures. The methods in the present manuscript need to be specified. If the methods have been described in other studies, please state briefly in the current paper.

Minor:

Line 101: will changeà changed

Figures: Please add full names for CK, LS, HS in every figure. It is expected that every figure should stand out independently, that said, readers don’t have to read text to understand the figures.

Line117: novo à de novo

Line119: Nine cDNA libraries: please explain what are their species treatments. Do you mean three replicates of each treatment?

Line133: Please give the full name and reference/link of these database.

Line 148: NR or Nr? Please make terms consistent.

Line117: I expected to see BUSCO results for transcriptome completeness. That’s very easy to run.

Line132: For unigenes or “assembly unigenes”, do you mean reference transcriptome? I think “reference transcriptome” is more clear than unigenes.

Line150-157: You don’t need to state these numbers in the text as they have been shown in the figure.

Line159-172: This paragraph looks like method or discussion. For results section, you only need to briefly introduced the method, please leave the details to the Method section.

Line204: Is the metabolite profiling targeted or not targeted?

Figure 8,9,10,11: Did you use the published graphs or pathways?  If so, please cite the references and acquire the agreement from authors of the original study for using the graphs.

Line 338: Discussion: It would be better if there is a paraph briefly summarizing the main results and significance of this study at the beginning of Discussion.

Line440: How did you measure PPFD with what tools?

Line451: Please show more details of ELISA and liquid chromatography.

Line457: Did you use isotope to measure WUE and nitrogen? Please specify details.

Line465: Please specify details of photosynthetic pigment determination. If it has been described in another article, please state briefly here.

Line475: The version and parameters of Trinity?

Line478-481: References for these databases?

Author Response

感谢您对我们的手稿提出意见。这些意见对我们论文的修改和完善都很有价值,对我们的研究具有重要的指导意义。我们仔细研究了评论并进行了更正,希望得到批准。 

Reviewer 2 Report

<Introduction:  must be improve.

I don't consider important the propagation part.

Third and fourth paragraph could be merge.

 <Results:

Line 131: I recommend explore the parameter Ex90N50, is more suitable than N50 in de novo. You could see it in: https://github.com/trinityrnaseq/trinityrnaseq/wiki/Transcriptome-Contig-Nx-and-ExN50-stats

Line 141: Which tool did they use to compare all the unigenes at the same time and asumed that?
Line 200: I consider more suitable the expression “the percentage of transcription factor” than “the statistical analysis”

Line 242: auxin misspelled

Figure 9: this image is from: https://doi.org/10.1371/journal.pone.0113782 . It should be mentioned so there is no conflict of interest. Legend of the heatmaps must be clear.

< Discussion:

It’s no mentioned the relation between Phytochromes and Shade avoidance. Do you check it?

< Material and methods:

Zeatin: maybe it should be mentioned more general, as cytokinin

4.4. Photosynthetic pigment determination -> It could be rewrite and expand the information

4.5 RNA extraction and Sequencing -> In NIH-NCBI it’s mentioned that reads was obtained by the equipment HiSeq TM 2500 but in the manuscript it’s mentioned HiSeq TM 2000. What’s the correct?

4.6. Analysis of differentially expressed genes (DEGs) and functional annotation -> I don't like the way is redacted at beginning, a possibility could be: "Gene expression levels for each gene were normalized using DESeq2 and summarized as FPKM (Fragments Per Kilo-base pair per Million reads) values"

4.7. Quantitative real-time polymerase chain reaction (qRT-PCR) validation  à I don’t see the RNA extraction, quantification and how it’s obtain the cDNA.

2-ΔΔCT method à No citation, consider: Winer, J., Jung, C. K. S., Shackel, I., & Williams, P. M. (1999). Development and validation of real-time quantitative reverse transcriptase–polymerase chain reaction for monitoring gene expression in cardiac myocytesin vitro. Analytical biochemistry, 270(1), 41-49. Also:    Schmittgen, T. D., Zakrajsek, B. A., Mills, A. G., Gorn, V., Singer, M. J., & Reed, M. W. (2000). Quantitative reverse transcription–polymerase chain reaction to study mRNA decay: comparison of endpoint and real-time methods. Analytical biochemistry, 285(2), 194-204.

4.8. Metabolite identification and functional analysis à line 501: a metabolome program was used (which?)

The explanation of metabolite extraction is too short, what about the temperature of the solvent? did they extract the supernant? I suggest this part should be re-redacted

Author Response

(The authors gave the same response as above.)
